# A Deep Learning Framework for Musical Acoustics Simulations

## Abstract

The acoustic modeling of musical instruments is a heavy computational process, often bound to the solution of complex systems of partial differential equations (PDEs). Numerical models can achieve a high level of accuracy, but they may take up to several hours to complete a full simulation, especially in the case of intricate musical mechanisms. The application of deep learning, and in particular of *neural operators* that learn mappings between function spaces, has the potential to revolutionize how acoustics PDEs are solved and noticeably speed up musical simulations. However, such operators require large datasets, capable of exemplifying the relationship between input parameters (excitation) and output solutions (acoustic wave propagation) per each target musical instrument/configuration. With this work, we present an open-access, open-source framework designed for the generation of numerical musical acoustics datasets and for the training/benchmarking of acoustics neural operators. We first describe the overall structure of the framework and the proposed data generation workflow. Then, we detail the first numerical models that were ported to the framework. Finally, we conclude by sharing some preliminary results obtained by means of training a state-of-the-art neural operator with a dataset generated via the framework. This work is a first step towards the gathering of a research community that focuses on deep learning applied to musical acoustics, and shares workflows and benchmarking tools.

## 1 Introduction

The study of the acoustics of musical instruments is a challenging topic. Physics phenomena underlying music making are quite various and include excitation, resonant behavior, as well as the coupling and the dynamic modification of the involved mechanical parts. These make musical instruments remarkable examples of engineering, but also acoustic systems difficult to model. The most accurate simulations that exist today leverage the numerical solution of partial differential equations (PDEs), that are in turn designed to model the specific acoustic behavior of the targeted instruments (Bilbao, 2009). Unfortunately, the majority of the employed solvers are characterized by heavy computational requirements, often leading to restrictive implementation conditions (e.g., low spatio-temporal resolution, high degree of model simplification, non-interactive paradigms).

Recent advancements in deep learning have shown how neural networks may be used to enhance and even replace traditional PDE solvers (Bhatnagar et al., 2019), with the aim to improve performance. In particular, the use of *neural operators* has yielded promising results in fluids dynamics (Li et al., 2020), suggesting that their application may be successfully extended to revolutionize the simulation of the acoustics and the aeroacoustics of musical instruments. Being completely data-driven, neural operators could be trained to solve acoustics PDEs with synthetic datasets, generated via the large array of traditional numerical implementations that are available in the literature[1].

Although exciting, this scenario is hindered by a lack of common practices that are needed to bridge the domains of musical acoustics and deep learning. These include shared datasets, benchmarks, as well as general tools to help researchers categorize, manage and employ acoustics data for training and inference. The aim of our research is to foster the rapid growth of an active community where

---

[1]In this scenario the only constraint would be computational time—an affordable caveat when generating training sets.

these common practices could be discussed and formalized, along with the overall emerging field of deep learning-based musical acoustics. In line with this mission, in this work we present the *Neuralacoustics* framework, a collection of open-access/open-source scripts and tools designed to address the aforementioned needs. In particular, we provide an in-depth description of the dataset generation workflow proposed as part of the framework, and we introduce the first numerical models available in it. We also discuss preliminary results obtained by training a state-of-the-art neural operator for the solution of a simple acoustics problem, using exclusively the tools available in the framework.

## 2 BACKGROUND

**Musical Acoustic Simulations.** In the musical domain, the practice of designing mathematical models of instruments is often referred to with the term *physical modeling synthesis*. Common techniques include modal synthesis (Causse et al., 2011) and digital waveguides (Smith, 1992). Yet, the most precise techniques rely on numerical analysis (Castagné & Cadoz, 2003) (e.g., finite elements, finite differences). Numerical models implement solvers of PDE systems; they can finely simulate fundamental aspects of musical acoustics, like wave propagation and aeroacoustics, as well as physical phenomena beyond instruments and music (Yokota et al., 2002; Arnela & Guasch, 2014). The downside of numerical approaches lies in the computational load of the resulting models, as well as in the amount of parameters they have to comprise to properly simulate the instrument's behavior.

Of particular interest to our work is the case of time-domain simulations of musical instruments (Bilbao, 2009). In this context, the PDEs solved by the models describe the relationship between previous and next states of the instruments, organized over discrete time steps. Other than taking into account time-varying acoustic excitation of the instruments, this approach potentially enables the design of interactive models. Despite the high computational requirements of numerical analysis, real-time interactive models of musical instruments have been designed in recent years (Sosnick & Hsu, 2010; Allen & Raghuvanshi, 2015; Zappi et al., 2017). Unfortunately, this approach relies on expensive dedicated hardware (GPUs) and implementations are characterized by noticeable technical constraints, that limit access to models' parameters and interaction (Renney et al., 2022). As a result, numerical analysis is employed for the greater part to model simple musical systems[2] (Bilbao et al., 2019), or for batch (i.e., non-real-time) simulations (Bilbao & Chick, 2013; Arnela & Guasch, 2014) that may require run-times of several hours. In both cases, the applicability as well as the intelligibility of the resulting models are heavily hindered.

**Deep Learning and PDE Solvers.** Recently, deep learning has been successfully explored for the generation of PDE solvers describing time-dependent problems (Blechschmidt & Ernst, 2021; Li et al., 2020). These neural solvers may reduce the overall computational requirements of traditional ones, while approximating their output with a remarkable degree of precision. One of the simplest examples of neural solvers consists of deep convolutional neural networks parametrizing the operator that maps inputs and outputs (i.e., solutions) of the PDEs (Bhatnagar et al., 2019; Khoo et al., 2021). The limitation to this approach lies in its dependence on the chosen mesh, meaning that it is not possible to compute solutions outside the discretization grid used for training. Physics informed neural networks solve this issue, as they are mesh-independent and designed to work alongside classical schemes (e.g., Runge-Kutta) (Raissi et al., 2019). Although capable of addressing problems in the small data setting and with high dimensionality (Blechschmidt & Ernst, 2021), they are often employed to solve time-dependent PDEs that share many similarities with the ones modeling musical acoustics—e.g., Navier-Stokes equations (Rudy et al., 2017; Font et al., 2021; Cai et al., 2022). Being only partially data-driven, this approach requires to tailor the network to a specific instance of the PDEs and to repeat training at any given new input.

Most of the individual advantages of the approaches introduced so far are collated in neural operators (Li et al., 2020). Neural operators are mesh-free operators that require no prior knowledge of the underlying PDEs. They learn mappings between infinite-dimensional spaces of functions relying only on a finite collection of observations; and they can be used without retraining to solve PDEs

---

[2]These numerical models can be deemed as "simple" only if compared to the complexity of actual acoustic instruments.

with different discretizations. Although very recent, they showed promising results not only in fluid dynamics (Li et al., 2020), but also in the solution of wave equations (Guan et al., 2021).

## 3  DEEP LEARNING AND MUSICAL ACOUSTICS

The application of deep learning to musical acoustics simulations is less straightforward than what it may seem. In the most general sense, the problem can be framed as mapping the state of a numerical model across the last $T_{in}$ time steps to its state across $T_{out}$ future time steps. Both convolutional neural networks (He et al., 2016; Ronneberger et al., 2015; Wang et al., 2020) and neural operators can be used to approximate this mapping with high degrees of confidence (Li et al., 2020), in most cases treating the input tensors like a time series of images/video frames. To this end, such networks have been trained by using numerical datasets that exemplify how the target acoustics model evolves over time given a set of initial conditions; and during inference, they can be used auto-regressively to generate a continuous output. These are promising results and stem from a working scenario that appears to align well with the general domain of acoustic problems.

However, these examples of PDE neural solvers do not take into account two important aspects that are specific to musical acoustic simulations. The first one pertains to the excitation of musical models. Rather than simply simulating the behavior of an instrument set into motion by an initial condition, acoustics numerical models can account for the effects of *continuous excitation* functions, that may drive the instrument throughout the full duration of the simulation. Examples of continuous excitations include basic sinusoidal waves, as well as gaussian pulses used to simulate mallet strikes on membranes and plates (Sosnick & Hsu, 2011), and glottal pulse trains that resonate in singing vocal tracts (Rosenberg, 1971; Guasch et al., 2016). In more advanced simulations, continuous excitation is not pre-computed; it is outputted by a self-oscillating system coupled with the main acoustics model, a common example being a reed coupled with the bore of a woodwind (Bilbao et al., 2015; Allen & Raghuvanshi, 2015). An excitation can start at any given time step of the simulation and the effects of consecutive/overlapping functions may be quite difficult to predict, especially in non-linear models. The training strategies explored so far in deep learning to predict the solution of time-dependent PDEs are not designed to capture this aspect of musical interaction.

A second aspect that is missing from the current state-of-art time-dependent PDE neural solvers is input/output heterogeneity (Figure 1). To allow for the synthesis of the instruments' sound, a numerical model has to output a field representing the physical quantity where the acoustic wave is propagating. A practical example is the model of a membrane that outputs how its displacement changes over time, with respect to its equilibrium position. In the simplest case, the output acoustic field represents the next state of the system. This means that, at any given time $t$, the latest output of the PDE becomes the input for the computation of the solution at time $t + 1$. This simple mapping can be approximated by networks characterized by feature and prediction tensors that represent the same single (acoustic) quantity, and that consequently allow for a straightforward auto-regression mechanism during inference. To our knowledge, this is the only working scenario explored in the literature on the application of deep learning to time-dependent PDEs (e.g., input/output flow velocity (Wang et al., 2020), input/output vorticity (Li et al., 2020)). Yet, in more complex numerical models the state of the system may be composed of more than one field, each representing a different physical quantity. This is required when the model implements an implicit solver, meaning that the next acoustic output is calculated via the solution of a coupled system of equations—hence the necessity for an extra state quantity. Other examples include models where the boundary conditions and/or the acoustic properties of the simulated materials vary over time, due to musical interaction (Bilbao et al., 2019; Zappi et al., 2017). We can refer to them as *acoustics parameters*.

Moreover, these two aspects of musical instruments' numerical modeling are often intertwined. While in some cases the excitation is directly applied to the field where wave propagation is simulated (and sampled for audio output), in many other models it is the additional state field that gets altered by the excitation function, only implicitly affecting the simulation output. This is the case for the simulation of woodwinds, where the state of the system is represented by both an acoustic pressure field and a flow velocity field, the latter carrying the velocity excitation incoming from the reed. In similar contexts, the input/output state mapping (including the excitation mechanism) is not trivial to approximate via a neural network, and the methodologies discussed in the literature cannot be applied in a straightforward manner.

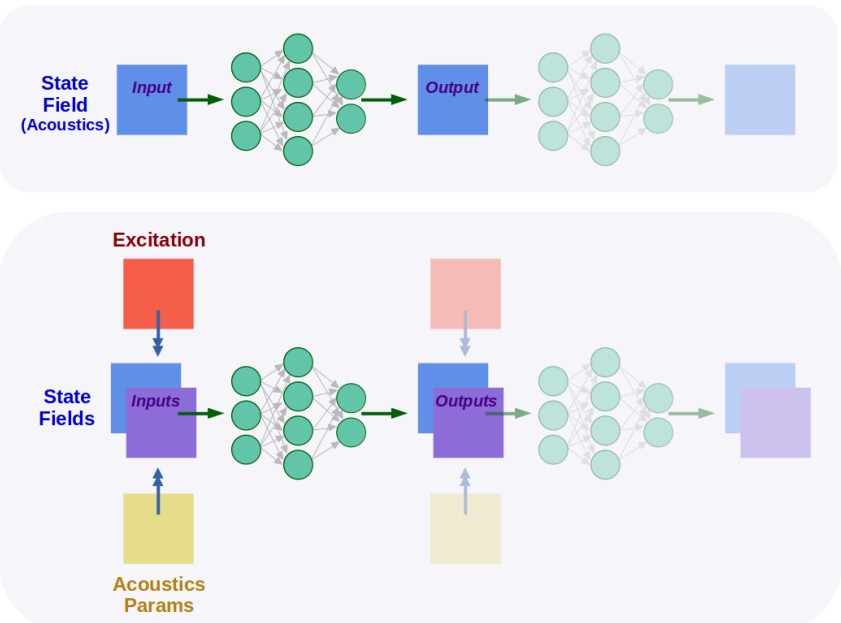

Figure 1: Neural network approximating the mapping between input/output states representing the same physical quantity (top); neural network approximating the mapping between heterogeneous input/output states, driven by continuous excitation and dynamic acoustics parameters (bottom). Only the first working scenario has been explored in the literature.

Novel network designs and training strategies may be explored that model these aspects of musical acoustics via deep learning. The fundamental requirement for the implementation of such algorithms is the availability of large datasets, that carry all the information needed to frame both the acoustic behavior of the simulated instrument and the inner workings of the solvers. This translates into storage of full state fields and excitations, along with standardized access methodologies for the extraction of data points as part of training sets.

## 4 DATASET GENERATION FRAMEWORK

The Neuralacoustics framework stems from the necessity to generate musical acoustics datasets that could be easily employed in deep learning. It consists of a collection of Python implementations of numerical models of musical instruments, embedded in a modular structure that facilitates extensibility and allows for the application of a model-independent workflow. Its overall structure and some of its features were inspired by the repository that in 2021 accompanied the work of Li et al. (2020). While building on this previous work, as discussed in the previous section we propose a framework that is specifically tailored to the case of musical instrument modeling and designed for extensibility. From a data-centric perspective, the design specifications of our framework adheres to the following constraints: the output of the acoustics simulations must be organized in data structures that are compatible with standard machine learning frameworks; such data structures must be easy to move between local and remote machines; and the output of each simulation must be easy to replicate.

The proposed framework can be accessed here[3]. It is written in Pytorch and requires the installation of a few additional libraries, mainly for the visualization of the acoustics simulations and for logging purposes. Pytorch was chosen for its flexibility and scalability on parallel architectures, yet the resulting datasets are not tied to this specific language (more details in Section 4.2). The dataset generation workflow that we propose acts upon three main types of components/scripts: solvers,

---

[3]https://anonymous.4open.science/r/neuralacoustics-737A/README.md

numerical models and dataset generators. In the following subsections, we detail these components and then we introduce the workflow in each of its steps.

## 4.1 Main Components of Synthetic Musical Acoustics Dataset Generation

**Solvers.** Solvers implement numerical solutions of highly parametric PDE systems, capable of modeling entire families of musical instruments. Regardless of the numerical method employed (e.g., finite difference, finite elements), All solvers have both a set of specific acoustics parameters, that depend on the implementation details, and a set of common parameters (e.g., domain size, simulation duration).

Currently, the framework includes three solvers, all based on finite-difference time-domain schemes. The first one was originally proposed by Adib (2000) and solves a PDE system capable of the simulation of damped transverse wave propagation in a two-dimensional medium. It can be used to model the basic acoustics of membranes, thin plates and rudimentary cymbals. The other two solvers tackle acoustic pressure propagation in 2D, and were ported from the OpenGL implementations proposed by Allen & Raghuvanshi (2015). The former is linear and it can be used to approximate woodwind bores; the latter includes non-linearities typical of brass instruments.

**Numerical Models.** Numerical models simulate specific musical instruments. To do so, each of these scripts loads a solver and sets some of its acoustics parameters, imposing for example constraints on the shape of the domain (i.e., spatial boundary conditions) or on the acoustic properties of the simulated materials. Furthermore, numerical models are characterized also by an excitation algorithm. Excitations can work as initial conditions simply aimed at setting the model into motion (e.g., initial displacement of a membrane), or as continuous models that excite the instrument throughout the whole duration of the simulation. By setting the parameters of the underlying PDEs and defining an excitation input, numerical models can be used to simulate not only different instruments, but also different playing configurations of the same instrument (e.g., a membrane hit by a stick or by a mallet). Each numerical model exposes controllable parameters. These include those pertaining to the excitation algorithm implemented in the script (e.g., the location of a hit, the area where an initial condition is applied), as well as any parameter of the solver that is not hard-coded by the model. These controllable parameters provide the ability to "tune" the behavior of the instrument and allow for the generation of datasets via the mechanism described in the next paragraphs.

For now, we included into the framework only four simple models, based on the aforementioned solvers. In particular, we implemented three models that simulate vibrating membranes, using as excitations impulses, noise and single spatial frequencies that match the horizontal modes of the surface, respectively. The fourth model leverages the linear acoustic pressure propagation solver to excite woodwinds bores with air wave pulses.

**Dataset Generators.** Dataset Generators consist of algorithms that load a specific model and automatically sample its parameters' space, effectively running large numbers of simulations of the same instrument/configuration. The framework includes two types of generators: random generators and grid-based generators. Random generators explore the parameters' space of the model using Python/Pytorch pseudo-random algorithms, driven by an arbitrary seed. This ensures determinism while avoiding clear patterns, and facilitates the reproducibility of results across different machines[4]. The number of random samples to take (i.e., the size of the resulting dataset) is passed to the generator as an input parameter. Grid-based generators do not rely on any random calculation, rather they sample the parameters' space in a linearly fashion. Each parameter to sample is assigned a range and a sampling step, then all the possible combinations of parameters are automatically computed—forming a "grid". Differently from the case of random generators, the total number of samples is not arbitrary but depends on the defined grid, and the resulting datasets need to be shuffled before being used for training.

Much like the case of numerical models, dataset generators expose a set of parameters. Example generator parameters include ranges and steps of the model's parameters to sample (for grid-based generators) and the requested number of dataset entries and the current seed (for random generators).

---

[4]We though noted that Pytorch deterministic algorithms may result in slightly different output numbers when the same generator is executed on different computer architectures.

## 4.2 DATASET GENERATION WORKFLOW

The main component scripts are not designed to be run directly. The framework includes instead an entry-level dataset generation script, that allows for the correct use of generators, models and solvers, and partially hides the complexity of the underlying code and dependencies. Once launched, this script collects all the simulation data computed by a chosen generator script into an actual dataset. More specifically, dataset entries represent complete simulations, each associated to a different set of parameters. In line with what discussed in Section 3, they consist of dictionaries containing all the inputs to the model (excitations and variable acoustics parameters[5]) and output solutions encompassing all state fields (rather than the acoustic output only), for every simulation time step.

The way in which datasets are structured and stored reflects the first two data-centric constraints we introduced at the beginning of this section, i.e., compatibility and portability. The entry-level dataset generation script outputs *MAT-files* (".mat" extension), one of the de-facto standard data formats in machine learning. The generation of acoustics datasets may yield very large files, especially when simulations span big domains and long time windows. To avoid exceeding the maximum size supported by the native file system where the code runs, the entry-level dataset generation script is capable of splitting the dataset into several "chunks", each represented by an individual MAT-file. This solution comes in handy also when moving datasets between remote locations, for the transfer of large files may be subject to failures due to connection instability. Eventually, when a dataset is loaded in memory (to train a network or to visualize its content), all the chunk files are transparently combined together back into a single dataset (more details in the next subsection).

The complete dataset generation workflow can be summarized as follows. First, the user has to locate a numerical model that represents the specific instrument the dataset will exemplify. Then, a dataset generator needs to be chosen, that samples the numerical model of interest. In this step, the user shall adjust the exposed parameters to make sure that the sampling procedure will result in data that well represent the instrument and its specific playing configuration[6]. Finally, the user can set and run the entry-level dataset generation script and the resulting dataset will be computed and stored according to the requested settings. The entry-level script compiles a log file too. It contains a summary of the content of the dataset and reports location and all parameters of the employed scripts. Any log file can then function as a unique configuration file, that when passed to the entry-level script allows users and remote collaborators to automatically obtain an exact copy of the original dataset, avoiding the hassle of moving large files or going through the full workflow again. The only caveat is that the same version of the framework has to be installed on both ends. This mechanism was designed to respect the third data constraint (replicability). Moreover, every step of the workflow can be carried out via command line, making it straightforward to check out the framework on remote machines and generate datasets on high performance computing clusters.

## 4.3 UTILITY TOOLS: VISUALIZATION AND DATA POINTS EXTRACTION

In the root of the framework, the user can find two further entry-level scripts, one designed to test numerical models, the other to visually inspect the content of datasets. Both scripts sequentially visualize the solution fields as plots, showcasing the propagating acoustic waves as an animation. The main difference between the two scripts is that the former computes the frames in real-time by means of running the tested model, while the latter extracts them from the inspected dataset. To this end, the dataset visualization script implements a windowing system that allows for the extraction of diverse sets of data points from the same dataset. To understand this mechanism, it is necessary to emphasize the difference between an "entry" within the dataset and a "data point" extracted from it. As detailed in Section 4.2, each entry in a Neuralacoustics dataset consists of a time series, representing the simulation of an instrument over a certain number of time steps. In the most general sense, a Neuralacoustics data point can be any sub-series of consecutive time steps found in a dataset entry. More than one data point can be extracted from a single dataset entry and the maximum size of a data point is equal to the total number of time steps of the simulation. In the latter case, only a single data point is extracted, which coincides with the full entry.

---

[5]Currently, all the models ported to the framework sport fixed acoustics properties, hence the dictionaries generated from these models do not include variable acoustics parameters as part of the input tensors.

[6]The specific metrics depends on the purpose and the application the dataset is computed for.

The windowing algorithm is part of a data point extraction tool, that retrieves data points by means of repeatedly windowing the entries' time series (collecting a data point per window). The process is depicted in Figure 2. The main windowing parameters are: the size of the windows, the stride applied between consecutive windows, and the dataset entry where the windows are applied. A further parameter allows to repeat the extraction over a number of consecutive entries, increasing the total number of frames visualized. To simply visualize the full simulations within each entry, the user can set either the size of the window equal to the number of time steps of each entry, or the stride equal to the size of the window.

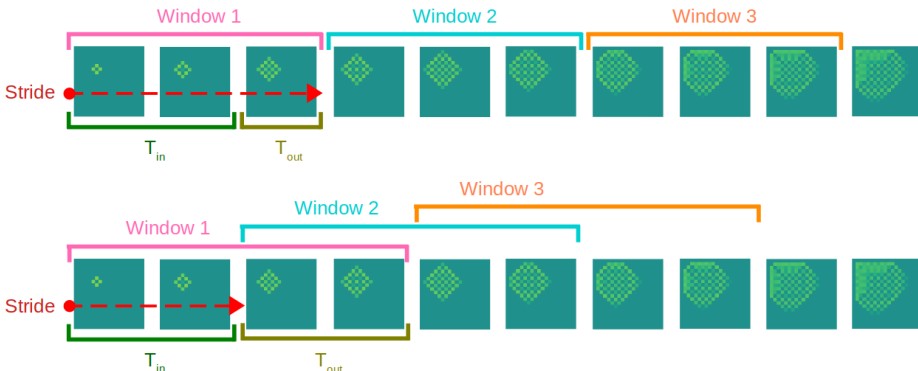

Figure 2: Windowing algorithm used to extract data points from Neuralacoustics datasets. The same dataset entry is processed via two different sets of parameters, to exemplify the extraction of different data. The diagram depicts only the solution time steps (acoustic fields) stored in the entry's dictionary.

When used to collate a training set, the extraction tool keeps applying the windows across each entry of the dataset, until the number of data points requested by the user is obtained. It is important to notice, though, that the values returned by the tool contain more information than simple feature vectors, for two reasons. First, the size of the window is defined as $T_{win} = T_{in} + T_{out}$; hence, the extracted values contain both the states pertaining to the initial $T_{in}$ time steps (i.e., the features) and the states pertaining to the following $T_{out}$ time steps (i.e., the label to predict). Second, as described in Sections 3 and 4.2, the values stored in each dataset entry may include heterogeneous inputs and outputs. When extracted, these pieces of data may be combined in different ways to represent the actual state of the model in each time step, depending on the specifics of the problem and of the solver. This means that some extra logic is necessary to define features and labels and be able to employ the data points for training. While increasing the overall complexity of the framework, this design detail makes data handling as generic as possible, allowing for the application of the same workflow even when very different numerical models/simulations/solvers and neural operators are in use. We share the details of a simple example of data handling logic for training in the next section.

## 5 TRAINING NEURAL OPERATORS

The Neuralacoustics framework also aims to facilitate the design of neural operators capable of solving acoustics PDEs. Currently, this part of the framework is comparatively less developed than the dataset generation workflow introduced in Section 4, but basic training and evaluation pipelines have been completed. Besides, preliminary training results have been obtained, which will be discussed in Section 5.1.

Similarly to numerical models, neural operators are implemented in Pytorch and expose hyperparameters. As of now, only a two-dimensional Fourier neural operator as introduced in (Li et al., 2020) is available in the framework. This was chosen as the first network to be ported into the framework because of its ability to solve time-based problems, as well as for its somewhat simple

internal structure[7]. The hyperparameters exposed by the port include spectral layer stack number, hidden size and Fourier transform modes. The network performs prediction one time step at a time and produces consecutive results in an auto-regressive manner.

In the root directory, the user can find a script that carries out the full network training pipeline. The user shall specify the training configuration, including: dataset and network selection, size of training and validation sets, data points extraction details (e.g., $T_{in}$ and $T_{out}$), learning and optimization parameters. This separation of training parameters and network hyperparameters ensures generalizability, potentially supporting the training of networks that diverge from the structure of the implemented Fourier neural operator. The training script operates on the dataset via the data point extraction tool described in Section 4.3. The logic used to combine inputs (excitations) and outputs (solution fields) into state tensors (i.e., definition of features and labels) is still hard-coded in the script. Per each dataset entry, features are defined as the first $T_{in}$ state tensors within the extracted time window, each computed as the sum between the current acoustic solution and the excitation at the next time step. This is the simplest way to embed continuous excitation into the input state of a numerical model. The label of each data entry consists of the last $T_{out}$ state tensors, simply defined as the acoustic solutions in those time steps. Once both data points and network are set, the script trains the neural model with the specified learning parameters. Weight checkpoints are also saved, using a customizable step interval.

Similarly to the data generation workflow, the training script compiles a log file that serves as a summary of the training details. All parameters related to the training process, alongside dataset generation parameters and network hyperparameters, are recorded so that the log file itself could be used as a unique configuration file for thoroughly replicating the neural model.

Currently an entry-level evaluation script provides intuitive insights into a trained network's performance. The users shall specify the network's checkpoint to evaluate, as well as the exact dataset and data entry for running inference on. This script visualizes the predicted domain state along with the ground truth acoustic solution (computed by the original numerical model and stored in the dataset) and their difference, for a chosen number of time steps. An example of visualization is provided in next section.

## 5.1 Preliminary Results

We trained a two-dimensional Fourier operator network on a dataset generated with the membrane model including impulse excitation as initial condition (Section 4.1, Numerical Models paragraph). Such a working scenario was chosen because the Fourier neural operator was originally not designed to learn mappings that include continuous excitation. We sampled the model with a generator that randomizes the location of the impulse across the domain (i.e., the area of the membrane), as well as the amplitude of the impulse. Each simulation spans 25 time steps. Both scripts are available in the framework. The width and the height of the 2D domain are both set to 64. The size of the training set is 25000 data points, while 5000 data points are used as validation set. Data points are extracted only from the first 20 time steps of each entry (this is a further setting available in the training workflow). The state of the model in each time step is chosen as the solution tensor in that step and the excitation tensors stored in the dictionary are ignored, except for the first one (initial condition). The input step number $T_{in}$ is 10 (the feature vector is composed of the previous 10 states) and the output time step $T_{out}$ is 1 (the label is the next state). The neural operator has 4 spectral layers with 12 Fourier modes within each layer and a hidden size of 20. We trained the network for 800 epochs with Adam optimizer and an initial learning rate of 0.001.

The final loss amounts to 0.0355. Figure 3 shows two snapshots of the visualization generated via the evaluation script. The test set extracted by the dataset is not limited to the first 20 time steps, but spans the full duration of the simulations (25 steps). As can be observed, the 2D domain's state outputted by the trained network is a close estimate of the ground truth simulated via the numerical solver. Moreover, through an auto-regressive inference approach, the trained network is able to carry out a good prediction of how an impulse propagates beyond the first 20 time steps exemplified in the training set. This suggests that the learning captured—at least in part—the underlying physical mechanisms of wave propagation.

---

[7]The source code of the Fourier neural operators is available at `https://github.com/zongyi-li/fourier_neural_operator`

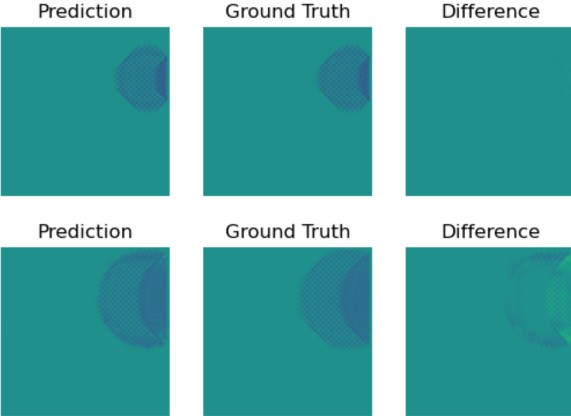

Figure 3: Evaluation visualization. From left to right, the three columns respectively shows the 2D domain state (displacement) predicted by the trained network, the ground truth, and their difference. The top row shows the states at time step #15 (exemplified in the training set), and the bottom row shows the states at time step #25 (never reached in the training set).

## 6 CONCLUSION AND FUTURE WORK

In this paper we introduced the Neuralacoustics framework, an open access and open source collection of tools designed to facilitate the application of deep learning in the context of acoustics simulations and musical instrument modeling. In particular, the framework responds to the need for standards to combine the output of diverse acoustics simulations into datasets, and to use them for training. The main components of the framework are numerical models and acoustic PDE solvers. These are arranged in a modular structure, that permits the application of a robust workflow for the generation of heterogeneous musical acoustics datasets. The generation process outputs data structures compatible with standard machine learning frameworks, and is designed to maximize portability and replicability. The Neuralacoustics framework features also a section dedicated to the training and the evaluation of neural operators using the generated acoustics datasets. While still in progress, this part of the workflow is functional and leverages a modular structure similar to the one proposed for the dataset generation process.

At the current stage of development, the framework includes Pytorch implementations of four numerical models and three solvers, all previously introduced in the literature. As showcased by the preliminary data presented in this work, these are enough to train a state-of-the-art neural operator and test its performance in the context of basic musical acoustics. Furthermore, the current implementations are designed to work as blueprints for the porting of additional models and solvers. With the release of this work, we aim to empower all the researchers working in this emerging field with new tools for the development and the sharing of their own implementations. The release of frameworks, common practices and benchmarks have long benefited the advancement of machine learning as well as its application in various domains (Downie et al., 2005; Schedl et al., 2014; Hu et al., 2020; Lu et al., 2021). We believe that our effort may have a similar impact on the development of novel deep learning approaches to acoustics modeling, and may facilitate the onset of collaborations among researchers from both fields.

The framework is constantly improved and extended. Besides the port of new numerical models (instruments) and neural networks/operators, we are currently working on three main fronts. The first one is quantitative evaluation on large datasets, with methods that in the near future will allow researchers to use the framework to precisely compare performance across multiple networks. The second one is direct performance comparison between neural and numerical models, including computational time/load and qualitative analysis. Finally, we are leveraging the framework to explore novel training strategies and deep learning architectures, designed to approximate the acoustics of more complex, interactive musical instruments' models.

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
