# OpenReview forum: "A Deep Learning Framework for Musical Acoustics Simulations"
_ICLR.cc/2023/Conference — Submitted to ICLR 2023_

### Official Review · Reviewer_HqQP · 2022-10-18

**Confidence:** 2
**Correctness:** 4
**Technical Novelty And Significance:** 3
**Empirical Novelty And Significance:** 2
**Recommendation:** 3

**Clarity, Quality, Novelty And Reproducibility:**

Reproducibility is clearly the highest. This paper is designed to be an open source framework and everything is described in great detail. Clarity and quality are also fine. The application of neural net PDE solvers to music acoustics is certainly novel and interesting

**Strength And Weaknesses:**

Strengths:
The problem itself is interesting, and using neural nets for PDE solving is a relatively new and exciting area. I have not seen neural networks applied specifically to PDEs for musical acoustics before. It is clear that great attention is paid to implementation details and good engineering practices. Those are listed clearly, and it should facilitate easy use by the community.

Weaknesses:
This paper is hard to judge because it focuses so much on the technical details of the implementation and not on the actual novel points and results. Firstly, the authors say that "The proposed framework can be accessed here" but the footnote says "Anonymized for Submission". Many sections detail out the file structure in great depth, but I cannot even see the framework or any of the points mentioned.   In addition, it is important to share qualitative results of musical instrument sounds that were simulated. What did they sound like, what kinds of instruments were generated and so on. This kind of novel framework could be valuable to the community. However, for this publication venue, the paper should focus less on the specific implementation details, file structure, etc.

**Summary Of The Paper:**

This paper introduces a new framework, called neuralacoustics, which uses neural networks to solve PDEs associated with the physical simulation of musical instrument sounds.

**Summary Of The Review:**

The idea is interesting and the framework could be valuable to the community. However, the paper does not actually include the framework or example results, and is focused far too much on implementation details for publication at this venue. I would suggest a more engineering focused venue, or to restructure the work to focus more on results and applicability to the community.

---

> ### Author Response · Authors · 2022-11-19
> **Response to Reviewer HqQP**
>
> We thank the reviewer for highlighting that our paper addresses a novel research topic where neural nets are applied to solve PDE in musical acoustics. We are happy to read that our intention of an easy-to-use, open source platform for this research topic is received well by the reviewer. We agree with the reviewer that some sections of the paper focus too much on the technical details of the implementation. As we mention in our replies to the other reviewers, we address this issue by taking out some technical details from the paper and moving them to the code repository (which is now available through a link to an anonymized GitHub repo—thanks for highlighting this issue). We then added a full section (Section 3) meant to highlight the novel points of our research. In particular, it discusses how the deep learning approaches presented in the literature so far cannot be directly applied to the case of musical acoustics, and how the proposed dataset generation framework may help researchers bridge these gaps. Finally, we do agree with the reviewer upon the importance of both quantitative and qualitative results. In this paper, it is our aim to present an overall workflow and detail the first numerical models/instruments that are now part of the framework, to exemplify its structure and scalability. As part of the next step of our research, we plan to run performance and perceptual comparisons between neural and numerical solvers, by means of simulating a variety of instruments. However, this is beyond the scope of the current submission. Here we provide a Research&Development environment where researchers from the ICLR community can start exploring neural acoustics and propose benchmarking tests, even when lacking strong knowledge of musical acoustics. Hopefully, this will in turn foster collaboration between experts in both deep learning and musical instruments. We thank the reviewer again for underlining that the contribution of the previous version of the paper was not clear enough and we hope that with this next iteration this issue is fixed.

---

### Official Review · Reviewer_iMqo · 2022-10-23

**Confidence:** 2
**Correctness:** 3
**Technical Novelty And Significance:** 2
**Empirical Novelty And Significance:** 1
**Recommendation:** 5

**Clarity, Quality, Novelty And Reproducibility:**

This paper provides explanations for their work. However, some important details are still missing.

**Strength And Weaknesses:**

I am not a person in acoustic areas. Thus, everything in this area should be new and exciting to me. However, from this description of this work, I just feel that what they are doing has been repeatedly considered by our researchers in image. Let me talk about some direct feelings about this work.
1. I have been writing a research proposal recently. This work looks like a research proposal. It does not look like a research paper. Why do you only provide preliminary results for your research paper? Does it mean that the authors have felt that their works have some substantial drawbacks needed to be considered in the future? If this work is just a preliminary result, why do you submit your work to a research conference?
2. Actually, I cannot find more detailed explanations for the dataset in this paper. Maybe, I missed some links. I just feel that if this is an open-access framework, is it possible to provide an external link to show some results or examples?
3. Besides, the method itself for data generation is not new. I have seen many descriptions of learning mappings between function spaces. If the authors just repeat what we have done for images in acoustic signals, I do not think it can ensure an ICLR paper. Maybe, ICASSP is better.
4. I read this paper for 1 hour. However, it looks like a technical report to me.

**Summary Of The Paper:**

This work introduces a framework for the acoustic modeling of musical instruments. The authors present an open-source and open-access framework for the generation of numerical muscial acoustics.

**Summary Of The Review:**

I have shown my concerns above. This paper just reads like a technical report.

---

> ### Author Response · Authors · 2022-11-19
> **Response to Reviewer iMqo**
>
> We would like to thank the reviewer for spending a lot of time carefully reading our paper. We agree with the reviewer that the paper included too many technical details, that shadow its research aspect. In response to that, we moved some of these details to the readme file of the repo. We then added a new section (Section 3) that discusses how the deep learning approaches used for images cannot be directly applied to the case of musical acoustics, and how the proposed dataset generation framework may help researchers bridge these gaps. Indeed, this is the main contribution of the paper, while the presented preliminary results are meant to exemplify the basic usage of the framework. As discussed in our answer to reviewer skE2, we hope that this set of tools may help deep learning researchers start the exploration of musical acoustics and work out new benchmarks and standards. We agree with the reviewer that the code repository link should be included in the paper. In the revised version, we added an anonymized GitHub repository link so that the reviewers can have a look at our code. Finally, our paper focuses on an interdisciplinary topic  that combines acoustic modelling with deep learning. We understand that the reviewer recommends a conference for audio/acoustics related topics. However, we believe we could receive a similar comment in response to  a submission to an audio/acoustics-focused conference, where the reviewer would recommend a deep learning conference as a better venue.  We believe that the design of the framework as well as the type of considerations presented in the paper are more suited to an audience carrying out research primarily in deep learning. As noted by the reviewer, this set of tools reminds of well known standards in ML, yet it is tailored to a somewhat unusual application. We are sure that this can ease deep learning researchers in musical acoustics, and possibly help the creation of interdisciplinary collaborations. Indeed, the framework is designed to allow for the development, testing and combined use of both deep learning and acoustics modeling code.

---

### Official Review · Reviewer_skE2 · 2022-10-25

**Confidence:** 2
**Correctness:** 3
**Technical Novelty And Significance:** 3
**Empirical Novelty And Significance:** 3
**Recommendation:** 5

**Clarity, Quality, Novelty And Reproducibility:**

The writing is clear, but the paper reads like a software API guide for the framework at times. The content presented is novel indeed.
The authors do plan to release their framework, hence the results shown in the work should be reproducible.

**Strength And Weaknesses:**

Strength:
- The authors identify acoustic simulation as a novel area of physics-based models where deep neural nets have not yet been widely applied.
- They develop a complete framework to facilitate deep learning-based research by implementing existing physical models in a familiar environment (Pytorch) for deep learning researchers. This is potentially very helpful for developing a new community in the research area.

Weaknesses:
- It is not clear to me how relevant this paper is for the community at ICLR. A large portion of this paper describes the framework and how it can be used to generate data using python implementations of numerical models of instruments. However, only a small part of the paper actually talks about modeling these numerical models with deep networks which may be more relevant to the community at large.
- Maybe this is obvious to others but I would be interested in seeing a comparison of the advantages/disadvantages of neural methods vs numerical methods for modeling the acoustics. For example, a user study comparing the outputs of two simulators, a comparison of simulation speed, etc.

**Summary Of The Paper:**

The paper discusses a new framework focusing on musical acoustics simulation. Traditional numerical methods for musical acoustics simulation involve solvers for partial differential equations to model the physics of musical instruments and sound propagation. These methods are very slow. The authors present the Neuralacoustics framework which implements various components necessary to support deep learning-based approaches to musical instrument simulation. It includes various solvers for numerical models which enable dataset generation which in turn can be used to train neural networks to approximate the numerical models. The authors provide a walkthrough of their framework and how to go about training a neural operator for a specific numerical model.

**Summary Of The Review:**

The paper describes a new framework for musical acoustics simulation which facilitates deep learning-based research. The paper goes over the various components in the framework followed by a case study of how it can be used for modeling a specific instrument type using neural operators. While I do not see any big issues in the technical details of the paper but I do find it difficult to gauge the interest in this work in the ICLR community. Hence I give the paper a rating of weak reject.

---

> ### Author Response · Authors · 2022-11-19
> **Response to Reviewer skE2**
>
> We thank the reviewer for their thoughtful comments. We are glad to read that the reviewer found several strengths in our paper. In regards to the weaknesses, we agree with the reviewer to add more details about the modeling of musical acoustics PDEs with deep learning. To this end, we added Section 3 that explains more in depth the challenges of this research topic as well as hopefully clarifies the value of the proposed framework to the ICLR community. Beyond this edit, we believe that the paper is relevant to the community. The breakthroughs that deep learning has already made in fluid simulations are exciting and we hope to see this type of advancement in musical instrument simulations too. Yet, musical acoustics is a much smaller niche compared to fluid dynamics and the odds of having expertise in both deep learning and instruments’ simulations are low. We think that the framework that we present in this paper can help deep learning researchers get involved in this exciting new application area, by providing an environment for benchmarking different deep learning architectures that does not require strong knowledge of musical acoustics modeling as entry fee. This would also facilitate the onset of interdisciplinary collaborations with musical acoustics researchers. Furthermore, we would like to emphasize the fundamental role that benchmarking has in ML. Example benchmarking efforts are common in Music Information Retrieval research communities, where the agreed benchmarking tasks have advanced the knowledge in the area in a fast and effective manner. Lastly, we agree with the reviewer that assessments including the analysis of audio outputs from different deep learning architectures or simulation speed comparisons with numerical implementations are very important. While these studies are in our future work list, the current paper’s objective is to provide a Research&Development environment where the research community can start exploring neural acoustics and propose benchmarking tests.

---

### Decision · Program_Chairs · 2023-01-20

**Decision:**

Reject

**Justification For Why Not Higher Score:**

The lack of qualitative results and the less focus on methodology and too much details on implementation make this paper not a good fit for this venue.

**Justification For Why Not Lower Score:**

N/A

**Metareview: Summary, Strengths And Weaknesses:**

Summary: The paper presents a new framework - Neuralaccoustics, which uses neural networks to solve PDEs associated with the physical simulation of musical instrument sounds.

Strength: the topic is interesting and the proposed framework can be potentially very helpful for developing a new research community.

Weakness: the paper presentation is more on what can be done, instead of how things can be done and there's no experimental justification on the feasibility. As pointed out by the reviewer that "the paper reads like a software API guide", "research proposal". Lacking of comparisons of advantages and disadvantages largely weakens the paper.